# Metatranscriptomic Analysis Reveals Rich Mycoviral Diversity in Three Major Fungal Pathogens of Rice

**DOI:** 10.3390/ijms23169192

**Published:** 2022-08-16

**Authors:** Zhenrui He, Xiaotong Huang, Yu Fan, Mei Yang, Erxun Zhou

**Affiliations:** Guangdong Province Key Laboratory of Microbial Signals and Disease Control, College of Plant Protection, South China Agricultural University, Guangzhou 510642, China

**Keywords:** mycovirus, metatranscriptomics, mycoviral diversity, rice fungal diseases, biocontrol

## Abstract

In recent years, three major fungal diseases of rice, i.e., rice blast, rice false smut, and rice-sheath blight, have caused serious worldwide rice-yield reductions and are threatening global food security. Mycoviruses are ubiquitous in almost all major groups of filamentous fungi, oomycetes, and yeasts. To reveal the mycoviral diversity in three major fungal pathogens of rice, we performed a metatranscriptomic analysis of 343 strains, representing the three major fungal pathogens of rice, *Pyricularia oryzae*, *Ustilaginoidea virens*, and *Rhizoctonia solani*, sampled in southern China. The analysis identified 682 contigs representing the partial or complete genomes of 68 mycoviruses, with 42 described for the first time. These mycoviruses showed affinity with eight distinct lineages: *Botourmiaviridae*, *Partitiviridae*, *Totiviridae*, *Chrysoviridae*, *Hypoviridae*, *Mitoviridae*, *Narnaviridae*, and *Polymycoviridae*. More than half (36/68, 52.9%) of the viral sequences were predicted to be members of the families *Narnaviridae* and *Botourmiaviridae*. The members of the family *Polymycoviridae* were also identified for the first time in the three major fungal pathogens of rice. These findings are of great significance for understanding the diversity, origin, and evolution of, as well as the relationship between, genome structures and functions of mycoviruses in three major fungal pathogens of rice.

## 1. Introduction

Rice (*Oryza sativa*) is one of the most important staple foods, and is a primary food source for more than half of the world’s population [1,2]. Rice is threatened by fungal pathogens at all stages of growth. In particular, large numbers of fungal pathogens have caused important diseases of rice, leading to widespread yield reductions and threatening global food security [3]. Rice blast, caused by the filamentous anamorphic fungus, *Pyricularia oryzae* (=*Magnaporthe oryzae*), is the most devastating disease of rice worldwide and is one of the 10 most important fungal diseases in plants [4,5]. Furthermore, rice-sheath blight, caused by the soil-borne fungus, *Rhizoctonia solani* Kühn, as well as rice false smut, caused by the notorious fungal pathogen, *Ustilaginoidea virens*, are emerging as important rice diseases globally [3,6,7]. Currently, however, there are no effective ways to defend against these fungal pathogens. It is widely accepted that breeding disease-resistant cultivars using disease-resistant genes from rice is the most economical and effective strategy for controlling the three major diseases of rice [8,9]. However, the complexity and variability of fungal pathogens have greatly limited the widespread use of many popular disease-resistant cultivars [5,9]. This prompted us to explore more efficient and eco-friendly strategies to defend against the fungal pathogens of rice.

Mycoviruses kinds of virus that infect fungi and are able to multiply in fungal cells [10,11]. In the last 60 years, since the first identification report of viruses infecting the cultivated button mushroom, *Agaricus bisporus*, knowledge of mycoviruses has grown exponentially [12]. A few mycoviruses, known as hypoviruses, can result in lower growth rates, decline of pathogenicity, and other phenotypes of host fungi. Therefore, hypovirulence-associated mycoviruses have potential for the biological control of plant fungal diseases. The successful biological control of chestnut blight, caused by *Cryphonectria parasitica*, with hypovirus-mediated hypovirulence, in the last century, is the best example of this and has inspired scientists to explore hypoviruses in more plant fungal pathogens [11]. Moreover, it has been reported that spraying Sclerotinia sclerotiorum hypovirulence-associated DNA virus 1 (SsHADV-1)-infected strain DT-8 at the early flowering stage can reduce the disease severity of rapeseed stem rot by 67.6% and improve yield by 14.9% [13]. Although most of mycoviruses are latent infections, some hypovirulence-associated mycoviruses have been reported in many fungi; as yet, they have not been tested on a large scale in the field.

As hypoviruses have potential for biological control and could also be applied to explore the mechanisms of interaction between the mycovirus and fungal host, it is obviously important to identify existing or potential mycoviruses in fungal pathogens to enable further characterization of their molecular biology and a risk assessment of their applications for biological control in the field. Since the first report of mycoviruses in 1962, virus-discovery studies have been performed in a variety of fungal species, including *Fusarium graminearum*, *Colletotrichum camelliae*, *Penicillium digitatum*, *Sclerotinia sclerotiorum*, etc. [14,15,16,17]. Initially, these studies utilized viral isolation, Sanger sequencing and cellulose powder to extract dsRNA, whereas, in recent years, more attention has been paid directly to metatranscriptomic sequencing [18,19,20]. Through metatranscriptomics, researchers have identified abundant mycovirus diversity in many widely dispersed plant-pathogenic fungi. In addition, metatranscriptomic sequencing has been applied to explore potential viruses in game animals. He et al. [21] recently reported that they identified 102 mammalian-infected viruses in 1941 game animals, with 65 described for the first time. In plant viruses, Sidharthan et al. [22] previously reported the identification of five putative novel poleroviruses and an enamovirus through plant transcriptomes.

Rice blast, rice false smut, and rice-sheath blight are the three most serious fungal diseases of rice, causing reductions in rice yield and quality. The current prevention and control measures of the three major fungal diseases of rice mainly rely on the cultivation of resistant varieties, the reduction in pathogen inocula, cultural control, and fungicide application. Fungal hypovirulent strains containing hypoviruses are attracting increasing attention because of their hypovirulent characteristics and potential application in the biological control of plant fungal diseases. However, there are few reports about mycoviral diversity in three major fungal pathogens of rice, especially hypovirulence-associated mycoviruses. To our knowledge, only Magnaporthe oryzae partitivirus 2 (MoPV2), Magnaporthe oryzae chrysovirus 1-A, -B, and -D (MoCV1-A, MoCV1-B, and MoCV1-D) can result in reduced growth rates, the decline of pathogenicity, and other phenotypes in the *P. oryzae* [23,24,25]. In *R. solani*, it was previously reported that Rhizoctonia solani partitivirus 5 (RsPV5), Rhizoctonia solani endornavirus 1 (RsEV1), Rhizoctonia solani partitivirus 2 (RsPV2), and Rhizoctonia solani dsRNA virus 5 (RsRV5) are hypovirulence-associated mycoviruses [26,27,28,29]. However, less hypoviruses have been reported in *U. virens*. To help fill this gap, we screened and cultured a total of 343 strains (118 from *P. oryzae*, 182 from *U. virens*, and 43 from *R. solani*) for metatranscriptomic sequencing in order to explore the diversity of mycoviruses in these three plant pathogenic fungi. Many of the mycoviral species were identified for the first time within a metatranscriptomic framework. Our goal was to reveal the mycoviral diversity and abundance in three major fungal pathogens of rice. The results of this study are of great significance for understanding the diversity, origin, and evolution of, as well as the relationship between, the genome structures and functions of mycoviruses in the three major fungal pathogens.

## 2. Results

### 2.1. Collection and Analysis of Three Major Fungal Pathogens of Rice

In previous studies, we collected and identified a large number of the three major fungal-pathogen samples of rice, mainly from the rice regions of southern China. The 118 *P. oryzae* strains were isolated from rice-blast samples collected in Hainan, Zhejiang, Guangdong, and Hunan provinces. The 182 *U. virens* strains were collected mainly from rice false smut samples in Hainan province. The 43 *R. solani* strains were collected from several provinces in southern China, including Hainan, Zhejiang, Hunan, Guangdong, Fujian, Guangxi, Jiangsu, Anhui, and Yunnan provinces (Figure 1a). These provinces are all areas in which the three major fungal diseases of rice are severely damaging. Thus, the fungal samples we collected and used in this study are broadly representative and potentially contain richer mycovirus diversity.

The complete sample collection comprised 343 strains from three fungal species, representing different filamentous fungal genera: *Pyricularia*, *Rhizoctonia*, and *Ustilaginoidea*. We performed a preliminary classification of these strains into normal and abnormal according to growth rate, colony morphology, and pathogenicity (Figure 1b). In the collected samples, some strains had significant hypovirulence, representing with biological characteristics such as slower growth rate, fewer sclerotia, abnormal pigmentation, abnormal colony morphology, and reduced pathogenicity (Figure 1c). The preliminary classification of the 343 tested strains showed that the abnormal strains accounted for 30.5%, 33.5%, and 25.6%, respectively, in the rice’s three major fungal pathogens (*P. oryzae*, *U. virens* and *R. solani*), which provided quality materials for mining mycoviruses.

### 2.2. Metatranscriptomic Identification of Mycoviruses Infecting the Tested Strains

To explore more enriched mycoviral information, we prepared a total of eight sequencing libraries using rRNA-depleted total RNAs from the three major fungal pathogens of rice (*P. oryzae*, *U. virens*, and *R. solani*). Through the metatranscriptomic sequencing of eight sequencing libraries, a total of 2.16 × 10^8^ raw reads were obtained in 118 *P. oryzae* strains, 2.88 × 10^8^ reads in 182 *U. virens* strains, and 7.82 × 10^7^ reads in 43 *R. solani* strains. These reads were de novo assembled into large contigs using the scaffolding contig algorithm, and 116,736 contigs, 106,131 contigs, and 99,590 contigs were obtained in *P. oryzae*, *U. virens*, and *R. solani*, respectively (Appendix A). Subsequently, all the contigs obtained were compared with the NR database, which is the most detailed protein database for the annotation of protein functions and structures at present. A total of 682 contigs were best matched with the viral genome. The results also showed that 79 contigs were obtained in *P. oryzae*, representing the partial genome segments of 22 mycoviruses, and 593 and 10 mycovirus-related contigs were obtained in *U. virens* and *R. solani*, respectively (Appendix A). We list the virus-related details in Table 1, including the provisional names, most closely related viruses, aa identity, length, etc. The analysis of the contigs assembled from the metatranscriptome data revealed that most of the sequence contigs related to mycoviruses showed low percentages of nucleotide and amino-acid sequence identities with known mycoviruses, and also included some known viruses. Through the BLASTn and BLASTx analysis, the putative viral genomes showed affinity with eight distinct lineages, including *Botourmiaviridae*, *Mitoviridae*, *Partitiviridae*, *Chrysoviridae*, *Hypoviridae*, *Narnaviridae*, *Polymycoviridae*, *Totiviridae*, and unclassified viruses (Figure 2a). In *P. oryzae*, the *Botourmiaviridae*-related viruses accounted for the largest percentage, of 45%, while in *U. virens*, the most dominant was the *Partitiviridae*-related viruses, with 32%. The putative mycoviruses were classified into double-stranded RNA (dsRNA) and positive-sense single-stranded RNA (+ssRNA) according to genome structure, with the most predominant type being dsRNA, accounting for 53% of the total viruses. Moreover, we found that the members of *Chrysoviridae* were detected only in *P. oryzae*, the members of *Hypoviridae* were detected only in *R. solani*, and the members of *Mitoviridae* and *Polymycoviridae* were present only in *U. vriens* (Figure 2b). In addition, we found that the majority of the colonies had normal morphology by observing the colony morphology of the virus-infected strains, which is consistent with previous reports that most mycovirus do not have significant effects on their hosts.

Further, we selected partial sequences to design specific primers for reverse transcription-polymerase chain reaction (RT-PCR) to validate the accuracy and reliability of the high-throughput sequencing. Based on the contig length, best-match species, and amino acid (aa) identity, we selected 97 contigs, representing different mycoviruses to design specific primers to verify the viral-origin strains (Figure 2c). The primer pairs used and the predicted sizes of the amplicons are listed in Appendix A. We focused on those novel mycoviruses, including viruses associated with hypovirulence and those not yet identified in the three major fungal pathogens of rice. The results of the RT-PCR validation showed that a total of 74 virus-related sequences from different viral families were identified. Viruses were identified in 66 of 108 abnormal strains and in 70 of 235 normal strains. A total of 219 virus-associated sequences were identified in these 66 abnormal strains, while a total of 137 virus-associated sequences were identified in these 70 normal strains. Compared to normal strains, abnormal strains contain more abundant viruses. Among the 136 virus-infected strains, 90 strains were co-infected by several viruses, while the *U. virens* strain 287 was co-infected by 9 different viruses, and this strain showed a significant hypovirulent characteristic. This suggests that the cause of the abnormal colony morphology may be viral infection. Furthermore, our results are consistent with previous reports that most viral infections are asymptomatic.

### 2.3. Sequences Related to Members of the Families Botourmiaviridae and Hypoviridae

According to the latest list of fungal viruses approved by the International Committee on the Taxonomy of Viruses (ICTV), the family *Botourmiaviridae* includes viruses infecting plants and filamentous fungi with +ssRNA genomes; each genomic RNA has one open reading frame (ORF). The family includes four genera, *Ourmiavirus*, *Botoulivirus*, *Magoulivirus*, and *Scleroulivirus*, of which the genome structures of *Botoulivirus*, *Magoulivirus*, and *Scleroulivirus* are monopartite, with a size of 2–3 kb, while *Ourmiavirus* is tripartate [30].

Among the tested strains, a total of 146 contigs showed similarity with mycoviruses in the family *Botourmiaviridae*. In total, 32 sequences were identified from *P. oryzae* and 114 from *U. virens*. According to statistics, 10 putative viruses that were identified as the members belonging to the family *Botourmiaviridae* were in the *P. oryzae* strains, and 6 viruses were in the *U. virens* strains. In *P. oryzae*, sequences of MoBV2A, MoBV4A, MoBV5A, MoBV6A, MoBV7A, MoBV9A, MoBV10, MoBV11, MoBV12, and MoBV13 were identified by BLAST to phylogenetically grouped families within the family *Botourmiaviridae* with segmented genomes. These sequences each contained a complete or partial ORF-encoding RNA-dependent RNA polymerase (RdRP). The products of these ORFs were associated with the RdRP proteins of the family *Botourmiaviridae*, which is a recently established viral family [31,32]. The amino-acid sequence identity of MoBV10, MoBV12, and MoBV13 with Epicoccum nigrum ourmia-like virus 1, Botrytis ourmia-like virus, and Sclerotinia sclerotiorum ourmia-like virus 2 were 85.3%, 31.3%, and 47.3%, respectively. In general, since the viruses in the family *Botourmiaviridae* have a genome size of about 2–3 kb, MoBV2A, MoBV4A, MoBV5A, MoBV7A, MoBV9A MoBV11, and MoBV13, with sizes of 2263 nt, 2092 nt, 3335 nt, 2227 nt, 2833 nt, 2539 nt, and 2209 nt, have a nearly complete genome sequence, but the sequences from MoBV6A, MoBV10, and MoBV12, with sizes of 1519 nt, 1180 nt, and 645 nt, respectively, representing only 21.5% to 76.0% of the genome, were less complete. In *U. virens*, we obtained 114 contigs, which may represent 6 novel members of the family *Botourmiaviridae*, which we named UvBV1, UvBV2, UvBV3, UvBV4, UvBV5, and UvBV6. The amino-acid sequences of the RdRP of the UvBV1 and UvBV2 were the most similar to the previously reported mycovirus within the genus *Magoulivirus*. The BLAST analysis suggested that UvBV3, UvBV5, and UvBV6 might represent novel viruses affiliated with the genus *Scleroulivirus*, and UvBV4 with the genus *Botoulivirus*. This is also the first identification of *Botourmiaviridae*-associated viruses in *U. virens*.

The *Hypoviridae*, is a family of cappedless viruses with positive-sense, single-stranded RNA genomes of 9.1–12.7 kb, possesseither a single large ORF or two ORFs [33]. Sclerotinia sclerotiorum hypovirus 2-5472 (SsHV2-5472) is a capsidless virus with +ssRNA genome of 14,581 nt, possessing a large ORF and affiliated with the family *Hypoviridae* [33]. Previously, many members of the family *Hypoviridae*, including Rhizoctonia solani hypovirus 1 (RsHV1), Rhizoctonia solani hypovirus 2 (RsHV2), and Rhizoctonia solani hypovirus 3 (RsHV3), have been identified in *R. solani* by metatranscriptomic sequencing, while no *Hypoviridae*-related viruses have been reported in *P. oryzae* and *U. virens* [18]. In this study, we identified a Contig 995 by metatranscriptomics sequencing, which was identified as being most similar to SsHV2-5472. It had an incomplete ORF, which encoded 203 aa, and BLASTp searching showed that this putative protein was almost identical with the polyprotein of SsHV2, with 100% identity. These results suggest that SsHV2 may be highly infectious and can naturally infect *R. solani*. Since some hypoviruses induce hypovirulence to host fungi, we can further explore the potential of using SsHV2 to control *R. solani*. A phylogenetic analysis of viruses in the families *Botourmiaviridae* and *Hypoviridae* produced similar results (Figure 3).

### 2.4. Narnaviridae- and Mitoviridae-Related Sequences

The members of the family *Nanaviridae* are the simplest RNA viruses, with +ssRNA genomes ranging from 2.3–3.6 kb, encoding only a single RdRP protein [34]. The family was once subdivided into two genera, i.e., *Narnavirus* and *Mitovirus*. According to the latest ICTV classification criteria, however, the family *Nanaviridae* contains only one genus, *Narnavirus*. We detected a total of four *Narnaviridae*-related contigs in the tested strains, one in *P. oryzae*, three in *U. virens*, and no related sequences were found in *R. solani*. Contig366 is 2531 nt, with an incomplete ORF encoding for 513 aa. The BLASTp analysis results showed this putative protein to be similar to the RdRP of Erysiphe-necator-associated narnavirus 35 with 71.8% identity. The phylogenetic analysis showed that Contig366 represents a sequence of the genus *Narnavirus*, and we named this novel virus Magnaporthe oryzae narnavirus 3 (MoNV3). In *U. virens*, we obtained a total of three *Narnaviridae*-related sequences, First_Contig93, Contig553, and Contig731, which are the first reports that narnavirus can infect *U. virens*. First_Contig93 is 1963 nt, containing one incomplete ORF-encoding RdRP protein with 526 aa. The BLASTp analysis showed that this protein was most similar to the RdRP of Botryosphaeria dothidea narnavirus 4, with 46.2% identity. Thus, First_Contig93 represented a novel narnavirus, which we named Ustilaginoidea virens narnavirus 1 (UvNV1). The sequence of Contig553 was 2395 nt, with one complete ORF, which encoded a 777 aa protein. The BLASTp analysis result showed that this putative protein had 67.2% identity with the RdRP of Plasmopara-viticola-lesion-associated narnavirus 43. The phylogenetic analysis also showed that the putative novel virus is affiliated with the genus *Narnavirus*, and we named it Ustilaginoidea virens narnavirus 2 (UvNV2). Contig 731 was 2398 nt, with a predicted amino-acid sequence most similar to the RdRP amino acid sequence of Sanya narnavirus 7, with 38.9% identity; we named it Ustilaginoidea virens narnavirus 3 (UvNV3). We also performed a phylogenetic analysis, and the results also indicated that these sequences are representative of novel members of the family *Nanaviridae*.

The predicted amino acid of one contig, First_Contig59, showed similarity with members of the family *Mitoviridae*. According to previous reports, members of the family *Mitoviridae* are naked +ssRNA viruses containing a single ORF-encoding RdRP, with a genome size of approximately 2.5–2.9 kb [35]. Mitoviruses infect fungi only, and more than 500 mitovirus-related sequences have been identified; however, this study is the first to report that mitovirus was found in the plant pathogenic fungus, *U. virens*. The First_Contig59 was nearly complete at 2570 nt from metatranscriptomic sequencing. The predicted amino-acid sequence of First_Contig59 was 47% identical to the RdRP of Sclerotinia sclerotiorum mitovirus 6 in the family *Mitoviridae*. Thus, we named this novel virus Ustilaginoidea virens mitovirus 1 (UsMV1). A sequence analysis showed that UsMV1 has a large ORF-encoding RdRP similar to those of members of the family *Mitoviridae*. A phylogenetic analysis of the RdRP of UsMV1 and other selected viruses showed that the novel virus clustered with members of the family *Mitoviridae* (Figure 4). This is the first time that a member of the family *Mitoviridae* has been identified in *U. virens*.

### 2.5. Sequences Related to Members of the Family Partitiviridae

Members of the family *Partitiviridae* have small, isometric, non-enveloped viruses with bisegmented dsRNA genomes of 3–4.8 kb [36]. It was previously reported that there are many mycoviruses within the family *Partitiviridae* in the three major pathogenic fungi of rice, such as Magnaporthe oryzae partitivirus 1 [37], Rhizoctonia solani dsRNA virus 3 [38], Ustilaginoidea virens partitivirus 2 [39], etc.

Five sequences similar to viruses within the family *Partitiviridae* were identified from *P. oryzae*. The Contig 49 with 1800 nt contained a complete ORF that encoded a putative protein with 539 aa, and with 88.9% identity to the product of RdRP of Magnaporthe oryzae partitivirus 3. The Contig 208 was 1592 nt and contained an incomplete ORF encoding the coat protein (CP) of 179 aa, which showed the greatest similarity with the Verticillium dahliae partitivirus 1 with 74.8% identity. Subsequently, we extended them to 1778 bp and 1593 bp by RACE, respectively, indicating that the initial assembly obtained near full-length sequences. Furthermore, since these two sequences always appeared in the same strain, we tentatively speculate that these two sequences constitute a partitivirus, although we need further verification. Contig673 was 1473 nt and had a complete ORF encoding putative CP with 420 aa. The amino-acid sequence of the predicted ORF product was identical to that of Magnaporthe oryzae partitivirus 1. Contig1078 was 1769 nt and contained a complete ORF encoding RdRP with 539 aa. The predicted amino-acid sequence of RdRP was almost identical to that of Magnaporthe oryzae partitivirus 2, with 99% identity. Contig 110 was 1563 nt, with a predicted amino-acid sequence most similar to the CP amino-acid sequence of Verticillium dahliae partitivirus 1, with 74.3% identity. We named the novel partitivirus as Magnaporthe oryzae partitivirus 5 (MoPV5).

Two viral genomes from *R. solani* showed similarity with viruses in the family *Partitiviridae*. Contig 989 and Contig 990 were 502 nt and 712 nt in length, respectively, and each contained an incomplete ORF encoding RdRP. The BLASTp analysis showed that the amino-acid sequences of the RdRP were identical to Rhizoctonia solani dsRNA virus 3 (RsRV3) and Rhizoctonia solani dsRNA virus 3, respectively. In a previous study, we obtained the complete sequences of Rhizoctonia solani partitivirus 2 (RsPV2) and Rhizoctonia solani dsRNA virus 3 (RsRV3), which are consistent with the present study. The complete nucleotide sequences of two dsRNA segments of RsPV2 were determined as 2020 bp (dsRNA-1) and 1790 bp (dsRNA-2) in length [29]. The RsRV3 genome consists of two segments of dsRNA (dsRNA-1, 1890 bp, and dsRNA-2, 1811 bp), each possessing a single ORF [38]. These results indicate that the transmission of the mycoviruses is stable and the accuracy of metatranscriptomic sequencing is excellent. In *U. virens*, we obtained 126 contigs possibly representing 13 partitiviruses. The amino-acid sequences of the RdRPs of UvPV4, UvPV5, UvPV7, and UvPV14 showed that they all had less than 80% similarity with known viruses, and we tentatively speculated that they might represent four novel members of the family *Partitiviridae*. The phylogenetic analysis produced similar results (Figure 5).

### 2.6. Twelve Predicted Novel Viruses in the Family Totiviridae

Members of the family *Totiviridae* have virions with a diameter of approximately 50 nm, and the virions contain a single molecule of linear uncapped dsRNA, 4.6–7.0 kbp in size, encoding the CP and RdRP proteins, respectively [10]. Many totiviruses have been discovered in the three major fungal pathogens of rice: Magnaporthe oryzae-virus 1 (MoV1), Ustilaginoidea virens RNA virus 1 (UvRV1), Ustilaginoidea virens RNA virus 4 (UvRV4), and Rhizoctonia solani totivirus 1 (RsTV1), among others [7,40].

A sequence, First_Contig297, was recovered from *P. oryzae* with similarity to the virus in the family *Totiviridae*. First_Contig297 is 5164 nt, with two large complete ORFs, ORF1 and ORF2, respectively. The BLASTp analysis showed that ORF1 encodes a CP protein with 788 aa and ORF2 encodes an RdRP protein with 830 aa. Both the predicted RdRP and the CP were similar to MoV2, with a strong identity, of 99%. The sequence-structure analysis showed that the termination codon of the ORF1 encoding the CP protein overlapped with the start codon of the ORF2 encoding RdRP in the tetranucleotide sequence AUGA, which is also a common feature of the members of the genus *Victorivirus* [10]. MoV2 was originally reported from the *P. oryzae* strain Ken 60–19, collected from Japan [41]. Subsequently, we also identified MoV2 in *P. oryzae* collected in southern China, suggesting that MoV2 is widely distributed and may have strong transmission ability.

In eleven contigs, Contig3788, First_Contig40, Contig94, Contig21259, First_Contig136, Contig28, First_Contig78, Contig392, First_Contig38, Second_Contig1, First_Contig22, the sequences likely represent the viral genome of novel totiviruses in *U. virens*. Four of the putative totivirus sequences from *U. virens* were at least 4.5 kb and, therefore, represented nearly complete sequences. The BLASTx analysis showed that First_Contig38 was 99.4% identical to Ustilaginoidea virens RNA virus M (UvRVM) and First_Contig22 was 96% identical to Ustilaginoidea virens RNA virus 6 (UvRV6), suggesting that we may have obtained new viral strains of UvRVM and UvRV6. The remaining nine contig sequences showed 36–82% identity with known viruses, which could represent novel virus sequences. The predicted amino-acid sequence from First_Contig27 was 49% identical to the Ustilaginoidea virens RNA virus 5, a member of the genus *Victorivirus*. The results of the BLASTp analysis revealed that the putative protein encoded by ORF1 was similar to the CP of Ustilaginoidea virens RNA virus 5 with 52% identity, and the putative protein of ORF2 was similar to the RdRP of Thelebolus microsporus totivirus 1, with 49% identity. We named this novel virus Ustilaginoidea virens RNA virus 7 (UvRV7). Contig3788 was 710 nt, with a predicted amino-acid sequence most similar to the RdRP amino acid sequence of Beauveria bassiana victorivirus 1, with 45% identity. We named the novel totivirus Ustilaginoidea virens RNA virus 8 (UvRV8). Similarly, we showed that First_Contig40, Contig94, Contig21259, First_Contig136, Contig28, First_Contig78, and Contig392 are new viruses that belong to the family *Totiviridae*, which we named Ustilaginoidea virens RNA virus 9 (UvRV9), Ustilaginoidea virens RNA virus 10 (UvRV10), Ustilaginoidea virens RNA virus 11 (UvRV11), Ustilaginoidea virens RNA virus 12 (UvRV12), Ustilaginoidea virens RNA virus 13 (UvRV13), Ustilaginoidea virens RNA virus 14 (UvRV14), and Ustilaginoidea virens RNA virus 15 (UvRV15), respectively.

A phylogenetic analysis based on multiple alignments of the RdRP amino-acid sequences of these novel viruses that belong to the family *Totiviridae* and other selected viruses was conducted, and results grouped the sequences into four well-supported distinct clades, suggesting that these viruses may be derived from different genera (Figure 6).

### 2.7. Chrysoviridae- and Polymycoviridae-Related Sequences

The members of the family *Chrysoviridae* with multi-segmented dsRNA genomes are small non-enveloped isometric viruses, approximately 40 nm in diameter [42]. Magnaporthe oryzae chrysovirus 1 (MoCV1) was the first member of the family *Chrysoviridae* to be identified in *P. oryzae*, and MoCV1 can sustainably impair growth and alter colony morphology in host cells [25]. To date, no members of the family *Chrysoviridae* have been identified in *U. virens* or *R. solani*. In this study, we identified a total of five sequences associated with the family *Chrysoviridae* in *P. oryzae*. Contig241, Contig500, Contig554, and Contig5875 were 3218 nt, 2861 nt, 2958 nt, and 2793 nt in length, respectively, and all contained a large, complete ORF encoding a putative protein. The BLASTp analysis showed that this putative protein was most similar to the hypothetical protein of MoCV1, with 91%, 97%, 96%, and 69% identity, respectively. Contig2839 with 3519 nt contained a complete ORF encoding a putative protein with 1127 aa; a BLASTp search showed this putative protein was most similar to the RdRp of MoCV1, with 98% identity. Altogether, these results suggest that an MoCV1-like virus may infect the *P. oryzae*, which we have named Magnaporthe oryzae chrysovirus 5 (MoCV5).

The members of the family *Polymycoviridae* possess multi-segmented and non-conventional encapsidated dsRNA genomes of approximately 7.5–12.5 kbp. According to current reports, polymycoviruses only infect fungi (ascomycetes and basidiomycetes) and oomycetes [17,43]. The family *Polymycoviridae* has only one genus, *Polymycovirus*. Typically, polymycoviruses have four genomic segments, although some have up to eight. Therefore, through metatranscriptomic sequencing, we identified five *Polymycoviridae*-related sequences, which might represent a potential novel member of genus *Polymycovirus*. A Contig 1709, which we named Ustilaginoidea virens polymycovirus 1 RNA1 (UvPMV1 RNA1), with 1057 nt, contained a complete ORF, which encoded a putative protein with 276 aa. A BLASTp search showed this putative protein was most similar to P4 protein of Setosphaeria turcica polymycovirus 2, with 64% identity. A Contig 6818, which we named Ustilaginoidea virens polymycovirus 1 RNA2 (UvPMV1 RNA2), with 1957 nt, contained a complete ORF, which encoded a putative protein with 630 aa. The predicted amino-acid sequence from UvPV1 RNA2 was 66% identical to the product of the putative methyltransferase of StPV2. Contig 7065 (named Ustilaginoidea virens polymycovirus 1 RNA3 (UvPMV1 RNA3)) and Contig 12,036 (named Ustilaginoidea virens polymycovirus 1 RNA4 (UvPMV1 RNA4)) are 2173 nt and 1475 nt, respectively, and contain a large complete ORF that encodes a putative protein. The predicted amino-acid sequence from UvPV1 RNA3 was 71% identical to the product of the hypothetical protein of StPV2, while the predicted amino-acid sequence from UvPMV1 RNA4 was 78% identical to the product of the putative RdRP of StPV2. Contig17803 was only 350 nt, with a putative incomplete ORF encoding a protein most similar to the RdRP of StPV2, with 59.4% identity. The phylogenetic analysis produced similar results (Figure 7). Taken together, the five contigs might represent a potential new member of the family *Polymycovirus*. According to previous reports, polymycovirus infection has been associated with decreased host growth and reduced pathogenicity [44]. Therefore, we will further explore the characteristics of the novel virus and its interaction mechanisms with hosts.

## 3. Discussion

We performed a large-scale survey of mycoviruses in three major fungal pathogens of rice, among which the *U. virens* virome was studied for the first time. This analysis identified a wide diversity of previously described and novel mycoviral species, some of which could result in reduced growth rates, declines in pathogenicity, and other phenotypes of their fungal hosts. In this study, we collected abundant strains of the three major fungal pathogens of rice in southern China, including the provinces of Hainan, Zhejiang, Hunan, Guangdong, Fujian, Guangxi, Jiangsu, Anhui, and Yunnan. These areas belong to the subtropical climate zone, with higher temperatures and humidity, making them suitable for microbial growth. Rice crops in these areas also suffer huge economic losses every year due to the infestation of fungal pathogens. Therefore, the strains we collected may have rich mycoviral diversity. However, although we collected strains of rice fungal pathogens from as many different areas as possible in this study, the sample sizes for the mycoviral diversity analysis were still relatively small considering the total rice-growing areas in China and worldwide. The ongoing surveillance of mycovirus diversity in the world’s major rice regions clearly remains of utmost importance for understanding the origin, spread, and evolution of mycoviruses.

Although most mycovirus infections do not have visible effects on their fungal hosts, a few hypovirulence-associated mycoviruses can result in reduced growth rates, declines in pathogenicity, and other phenotypes of their host fungi, which have the potential for the biological control of plant fungal diseases. In this study, we performed a preliminary classification of the collected strains based on growth rate, pathogenicity, and colony morphology, and we found that 31.5% (108/343) of the strains had hypovirulent characteristics. According to previous reports, a few hypoviruses have been identified in the three major fungal pathogens of rice. For example, MoCV1 belongs to the family *Chrysoviridae*, which not only impairs host growth, but also results in abnormal pigmentation [25]. Our laboratory previously reported that the infections of *R. solani* by RsEV1, RsPV2, and RsRV5 resulted in reduced mycelial growth and hypovirulence to rice leaves [27,28,29]. However, no hypovirulence-associated mycoviruses have been reported in *U. virens* so far. In this study, we identified 73 mycovirus-related sequences by RT-PCR in three major fungal pathogens of rice. Combining the results of the preliminary biological experiments, we identified 219 virus-related sequences in 108 abnormal strains. Statistically, 66% (90/136) of the virus-infected strains were co-infected with several viruses. Therefore, we speculate that viral infection is likely to have been the main reason for the abnormal colony morphology of these strains. It is highly likely that the fungal strains tested in this study contained some hypoviruses, which will have served as the potential biocontrol resources for controlling the three major fungal diseases of the rice. In addition, we also identified viruses in 70 normal strains, which suggests that these viral infections do not alter the biological characteristics of fungal hosts. In the subsequent exploration, we will focus more closely on these hypovirulence-associated mycoviruses.

With the growing application of metatranscriptomic sequencing technology in mycoviral diversity research, a great number of novel viruses have been discovered and identified, and this has supported the progress of research in their related fields, such as virus pathogenesis, the control of related diseases, and so on. Due to the limitations of detection technology, we previously identified only dsRNA viruses in fungi, but with the widespread availability of high-throughput sequencing technology, increasingly novel +ssRNA, negative-sense single-stranded RNA (-ssRNA) and single-stranded DNA (ssDNA) viruses are now being identified. According to the classification of viral genome types, we have identified 32 sequence-associated +ssRNA viruses, accounting for 32.4% of the total viral sequences, in contrast to our previous understanding of mycoviruses as dsRNA viruses. Because previous research methods identified viruses by extracting dsRNA from fungal strains, this approach would have led us to find viruses that were primarily dsRNA. Therefore, metatranscriptomic sequencing has significant advantages and is widely utilized in mycovirus research.

The advances in metatranscriptomic sequencing technology facilitate the identification of known and unreported mycoviruses in the target fungi [19,45]. In this study, we identified sequences that putatively represented 68 fungal virus genome segments, some of which represent almost the full lengths of the viruses, while others were shorter sequences. The analysis also expanded the types of viruses associated with plant pathogenic fungi. In *P. oryzae*, we identified four sequences, MoBV6, MoPV1, MoPV2, and MoV2, with strong identity (>98%) with known viruses, and the remaining sequences may all represent novel viruses or new viral strains. In *U. virens*, we identified the members of the families *Botourmiaviridae*, *Mitoviridae*, *Narnaviridae*, and *Polymycoviridae* for the first time, indicating that a large number of viruses remain unexplored in *U. virens*. Previous reports suggested that the members of the families *Polymycoviridae* and *Mitoviridae* could result in the abnormal growth and declining pathogenicity of host fungi. Polymycovirus infection has been associated with various morphological alterations in host cells, including changes in pigmentation and sectoring [20,46]. Mitoviruses were previously reported to infect fungi only, but, more recently, it has been shown that a mitovirus, Chenopodium quinoa mitovirus 1 (CqMV1), can infect plants and damage them [47]. Therefore, we speculate that the polymycoviruses and mitoviruses in *U. virens* are highly probable hypovirulence-associated mycoviruses, and their biological characteristics need to be further explored. In *R. solani*, we found many previously reported mycoviruses in the tested strains, indicating the widely and prevalently distributed nature of the mycoviruses. We also identified a Sclerotinia sclerotiorum hypovirus 2 (SsHV2) in *R. solani*, which was previously identified in *Sclerotinia sclerotiorum*, suggesting that SsHV2 can cross-border infect *R. solani*. The question of whether SsHV2 can cause the abnormal growth of both *R. solani* and *S. sclerotiorum* stimulated our research interest.

Co-infection is a common phenomenon, in which two or more viruses simultaneously infect the same fungus under natural conditions, and it has been found in many fungi [48]. It was previously reported that a strain of *S. sclerotiorum* was infected by nine viruses, and the nine mycoviruses were identified and assigned to eight potential families [49]. Co-infection has also been reported in *Purpureocillium lilacinum* and *Beauveria bassiana*, two important entomopathogenic fungi, as well as many other fungal pathogens [50,51]. In this study, we also found the prevalence of co-infection by RT-PCR in the tested strains. In the co-infection of plant viruses, it has been reported that one virus has a significant effect on the replication or the transmission of a second virus [52]. For example, after infection with *Rice black-streaked dwarf virus* (RBSDV), rice plants are more resistant to *Southern rice black-streaked dwarf virus* (SRBSDV), but more susceptible to *Rice stripe virus* (RSV) [52]. In subsequent experiments, we intend to obtain derivative strains with different combinations of viruses through protoplast regeneration to explore the relationships between different viruses.

Further, the mining of rich mycoviral diversity could also be used to explore the interaction mechanism between mycovirus and its host fungus. Fusarium graminearum virus 1 (FgV1) could carry out an effective defense strategy by interfering with host-RNA silencing [53]. Hexagonal peroxisome (Hex1) protein is a fungal protein that can affect the accumulation of FgV1 RNA in host fungal cells [54]. A metabolomic analysis showed that the RsEV1 infection of *R. solani* resulted in the differential expression of 32 metabolites in *R. solani*, which are mainly involved in the pentose and glucuronate interconversions and glyoxylate, dicarboxylate, starch, and sucrose metabolism, among others [28]. The exploration of the mechanism through which hypovirus results in reduced pathogenicity to hosts will provide new insights into the application of mycoviruses in the biocontrol of plant fungal diseases. However, among the three major fungal pathogens of rice, relatively little research has been conducted on the hypovirulence-associated mechanisms of mycoviruses. In this study, we characterized a large number of novel mycovirus genomic sequences that can facilitate the deciphering of mycovirus–host-fungus interactions at the molecular level.

## 4. Materials and Methods

### 4.1. Fungal Strains, Cultural Conditions, and RNA Extraction

The 343 fungal strains used in this study were originally isolated from diseased rice samples collected in several provinces of southern China, including 118 *P. oryzae* strains, 182 *U. virens* strains, and 43 *R. solani* strains (Appendix A). The *P. oryzae* and *R. solani* strains were grown on potato dextrose agar (PDA) plates at 26 °C, while *U. virens* were maintained on potato sucrose agar (PSA) plates. These strains were maintained on PDA slants at 4 °C and on filter paper at −20 °C throughout the study. In order to obtain more detailed virome-characterization information, we divided the 118 *P. oryzae* strains into three groups, with 39 strains in each group, except group 3, which had 40 strains, and the 182 *U. virens* strains were divided into four groups, while the 43 *R. solani* strains were divided into one group. We grouped the strains according to their biological characteristics and geographical origins, and each group contained both normal and abnormal strains, as well as the strains from different geographical origins. Metatranscriptomic sequencing was performed separately for each group of strains. To extract the total RNA, all the strains were cultured on a cellophane membrane overlaying a PDA or PSA plate. The mycelia from different strains in the same group were mixed and ground with a mortar and pestle into fine powder in the presence of liquid nitrogen. The total RNA was extracted from each group using a TransZol Up Plus RNA Kit (TransGen Biotech, Beijing, China), following the manufacturer’s instructions, and checked for a RIN number to inspect RNA integrity by an Agilent Bioanalyzer 2100 (Agilent technologies, Santa Clara, CA, USA). Qualified total RNA was further purified by RNAClean XP Kit (Beckman Coulter, Inc, Kraemer Boulevard Brea, CA, USA) and RNase-Free DNase Set (QIAGEN, GmBH, Hilden, Germany).

### 4.2. Metatranscriptomic Sequencing and Bioinformatic Analysis

The extracted total RNA from each group was used to prepare individual sequencing library after the removal of rRNA and sequenced on the Illumina HiSeq 2500 by the Shanghai Biotechnology Corporation. The library construction was performed using Truseq Stranded Total RNA Library Prep Gold (Illumina, 20020598), and a total of 8 libraries were constructed in this study. The Raw Reads obtained from sequencing contained unqualified reads; we filtered them through the following main filtering steps to obtain Clean Reads that can be used for data analysis. The unqualified reads were filtered out, and they contained low-quality scores (<20) in the raw data, adaptor sequences, reads of less than 20 bp, and host-mRNA and -rRNA sequences. Next, de novo assembly was performed using the scaffolding contig algorithm of CLC Genomics Workbench (version:6.0.4), with word-size = 45, minimum contig length >200. The sequences obtained at this stage are called primary UniGene. Primary UniGenes were then spliced with CAP3 EST to construct the First_Contig and Second_Contig. The contigs obtained were then subjected to BLAST against GenBank using BLASTn and BLASTx. From the BLAST results, we obtained information such as best-matched species, aa identity, and classification. Our sequence data are available for download from the Sequence Read Archive, with accession no. PRJNA845944.

### 4.3. Putative Mycovirus Sequence Confirmation

According to the assembled contigs, we designed specific primers to verify which strains contained putative mycoviruses (see Appendix A for details). The RNAs were extracted from each strain and used as templates for complementary DNA (cDNA) synthesis using EasyScript^®^ One-Step gDNA Removal and cDNA Synthesis SuperMix (TransGen Biotech, Beijing, China), according to the manufacturer’s instructions. It is worth noting that during the synthesis of cDNAs, we need to select random primers. Subsequently, the presence of viruses was verified by RT-PCR using original RNA samples with species specific primers.

### 4.4. Phylogenetic Analysis

We selected assembled contigs that were high in identity with known viral nucleic acids and proteins for phylogenetic analysis. Meanwhile, the top 10 viral sequences showing sequence identity with the assembled contigs were retrieved in National Center for Biotechnology Information (NCBI). To analyze the relationship between the assembled contigs and known viruses, Clustal W in MEGA11 was used for alignment; next, phylogenetic trees were constructed with the maximum-likelihood method with a bootstrap value of 1000 replicates through MEGA11.

## Figures and Tables

**Figure 1 ijms-23-09192-f001:**
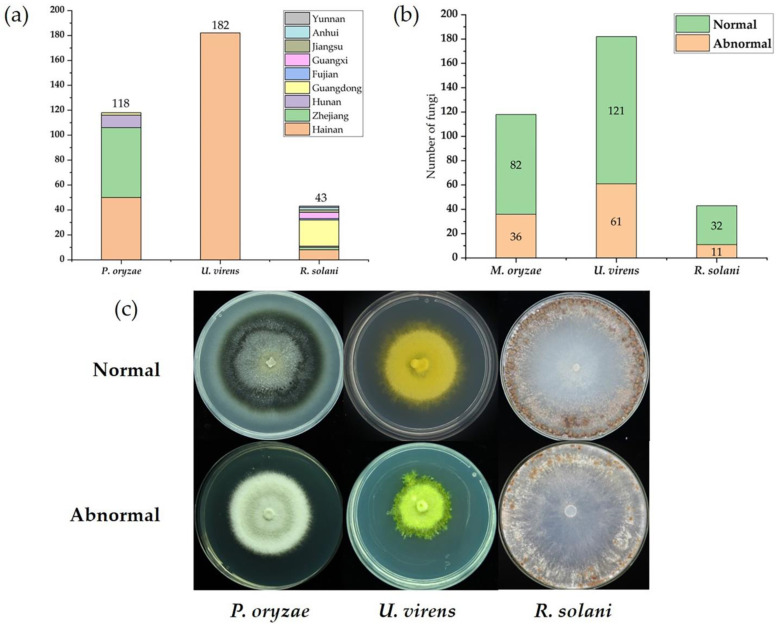
Fungal pathogens (*Pyricularia oryzae*, *Ustilaginoidea virens*, and *Rhizoctonia solani*) analyzed in this study. (**a**) Distribution of fungal samples by sampling provinces. (**b**) Classification of fungal strains by biological characteristics. (**c**) The colony morphologies of some representative normal and abnormal fungi used in this study.

**Figure 2 ijms-23-09192-f002:**
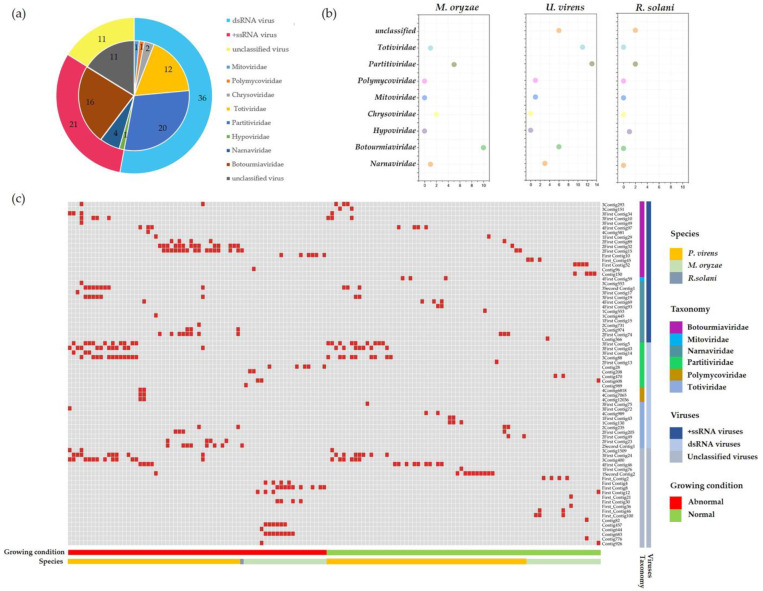
Overview of the virome in the three major fungal pathogens (*P. oryzae*, *U. virens*, and *R. solani*) of rice. (**a**) Numbers of mycoviral species in each viral family. (**b**) Prevalence of each viral family in the fungal species tested. (**c**) Distribution of viruses in the three major fungal pathogens of rice. The growth status of the strains was recorded at the time of culturing and reflected in the corresponding colors. Viral species from different viral families are shown, with each viral family indicated by the colors on the heatmap.

**Figure 3 ijms-23-09192-f003:**
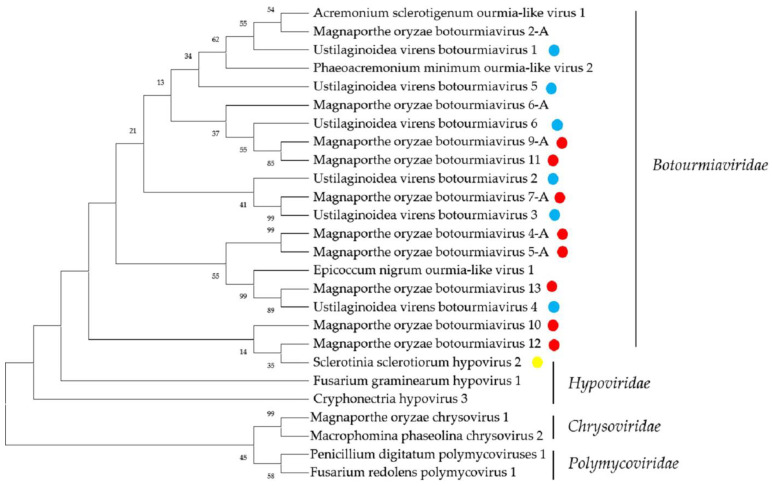
Phylogenetic analysis of viruses in the families *Botourmiaviridae* and *Hypoviridae*. Neighbor-joining-method phylogenetic tree based on the core RNA-dependent RNA polymerase (RdRP) motifs of putative viruses in the families *Botourmiaviridae* and *Hypoviridae*, other related viruses, and representatives of the two families. Viruses marked with red color are found in *P. oryzae*, blue color in *U. virens*, and yellow color in *R. solani*. The viruses without a color annotation are the representatives of the known species of each viral family.

**Figure 4 ijms-23-09192-f004:**
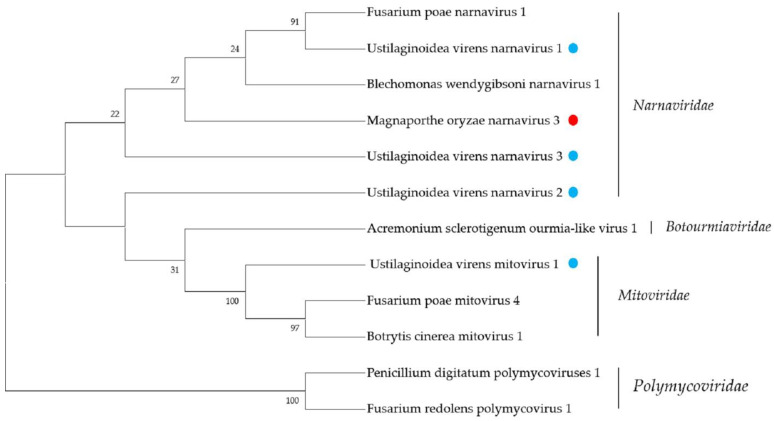
Phylogenetic analysis of viruses in the families *Narnaviridae* and *Mitoviridae*. Neighbor-joining-method phylogenetic tree based on the core RdRP motifs of putative viruses in the families *Narnaviridae* and *Mitoviridae*, other related viruses, and representatives of the two families. Viruses marked with red color are found in *P. oryzae* and blue color in *U. virens*. The viruses without a color annotation are the representatives of the known species of each viral family.

**Figure 5 ijms-23-09192-f005:**
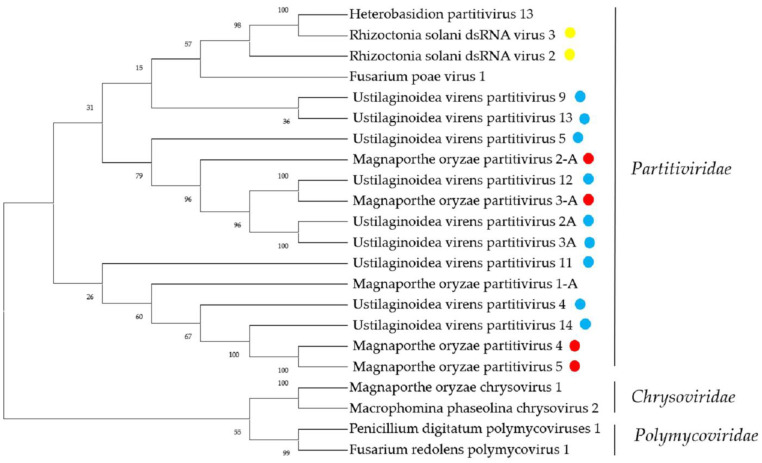
Phylogenetic analysis of viruses in the family *Partitiviridae*. Neighbor-joining-method phylogenetic tree based on the core RdRP motifs of putative viruses in the family *Partitiviridae*, other related viruses, and representatives of the family. Viruses marked with red color are found in *P. oryzae*, blue color in *U. virens*, and yellow color in *R. solani*. The viruses without a color annotation are the representatives of the known species of each viral family.

**Figure 6 ijms-23-09192-f006:**
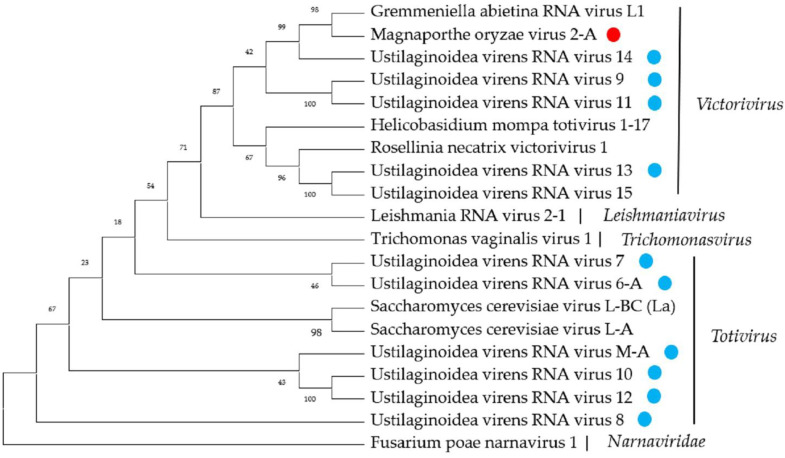
Phylogenetic analysis of viruses in the family *Totiviridae*. Neighbor-joining-method phylogenetic tree based on the core RdRP motifs of putative viruses in the family *Totiviridae*, other related viruses, and representatives of the family. Viruses marked with red color are found in *P. oryzae* and blue color in *U. virens*. The viruses without a color annotation are the representatives of the known species of each viral family.

**Figure 7 ijms-23-09192-f007:**
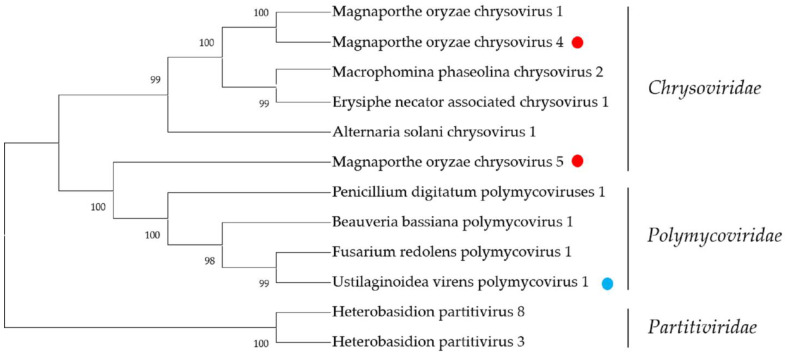
Phylogenetic analysis of viruses in the families *Chrysoviridae* and *Polymycoviridae*. Neighbor-joining-method phylogenetic tree based on the core RdRP motifs of putative viruses in the family *Chrysoviridae* and *Polymycoviridae*, other related viruses, and representatives of the two families. Viruses marked with red color are found in *P. oryzae* and blue color in *U. virens*. The viruses without a color annotation are the representatives of the known species of each viral family.

**Table 1 ijms-23-09192-t001:** Assembled sequences with similarity to those of previously described viruses.

Name of Putative Viruses	Contig Number	Accesson	Length	Best Match	aa Identity	Family
Magnaporthe oryzae narnavirus 3(MoNV3)	Contig366	ON791620	2531	Alternaria tenuissima narnavirus 1	72.864	*Narnaviridae*
Magnaporthe oryzae botourmiavirus 6-A(MoBV6A)	Contig2016	ON791624	1519	Magnaporthe oryzae botourmiavirus 6	98.683	*Botourmiaviridae*
Magnaporthe oryzae botourmiavirus 6-A(MoBV6A)	Contig2016	ON791624	1519	Magnaporthe oryzae botourmiavirus 6	98.683	*Botourmiaviridae*
Magnaporthe oryzae botourmiavirus 4-A(MoBV4A)	First_Contig6	ON791622	2092	Magnaporthe oryzae ourmia-like virus 4	96.853	*Botourmiaviridae*
Magnaporthe oryzae botourmiavirus 5-A(MoBV5A)	First_Contig10	ON791623	3335	Magnaporthe oryzae botourmiavirus 5	96.013	*Botourmiaviridae*
Magnaporthe oryzae botourmiavirus 9-A(MoBV9A)	Contig527	ON791626	2833	Magnaporthe oryzae botourmiavirus 9	95.302	*Botourmiaviridae*
Magnaporthe oryzae botourmiavirus 7-A(MoBV7A)	First_Contig140	ON791625	2227	Magnaporthe oryzae botourmiavirus 7	94.545	*Botourmiaviridae*
Magnaporthe oryzae botourmiavirus 10(MoBV10)	Contig66	ON791628	1180	Epicoccum nigrum ourmia-like virus 1	85.294	*Botourmiaviridae*
Magnaporthe oryzae botourmiavirus 11(MoBV11)	Contig150	ON791627	2539	Pyricularia oryzae ourmia-like virus 3	79.149	*Botourmiaviridae*
Magnaporthe oryzae botourmiavirus 13(MoBV13)	Contig75	ON791630	2209	Sclerotinia sclerotiorum ourmia-like virus 2	47.3	*Botourmiaviridae*
Magnaporthe oryzae botourmiavirus 12(MoBV12)	Contig102	ON791629	645	Botrytis ourmia-like virus	31.3	*Botourmiaviridae*
Magnaporthe oryzae chrysovirus 4(MoCV4)	Contig2839	ON791617	3519	Magnaporthe oryzae chrysovirus 1	91.408	*Chrysoviridae*
Magnaporthe oryzae chrysovirus 5(MoCV5)	Contig5875	ON791618	2793	Magnaporthe oryzae chrysovirus 3	58.9	*Chrysoviridae*
Magnaporthe oryzae partitivirus 2-A(MoPV2A)	Contig1078	ON791632	1769	Magnaporthe oryzae partitivirus 2	99.545	*Partitiviridae*
Magnaporthe oryzae partitivirus 1-A(MoPV1A)	Contig673	ON791631	1473	Magnaporthe oryzae partitivirus 1	99.456	*Partitiviridae*
Magnaporthe oryzae partitivirus 5(MoPV5)	Contig110	ON791635	1563	Penicillium aurantiogriseum partitivirus 1	91.358	*Partitiviridae*
Magnaporthe oryzae partitivirus 3-A(MoPV3A)	Contig49	ON791633	1800	Magnaporthe oryzae partitivirus 3	86.697	*Partitiviridae*
Magnaporthe oryzae partitivirus 4(MoPV4)	Contig208	ON791634	1592	Verticillium dahliae partitivirus 1	73.2	*Partitiviridae*
Magnaporthe oryzae virus 2-A(MoV2A)	First_Contig297	ON791619	5164	Magnaporthe oryzae virus 2	99.167	*Totiviridae*
Magnaporthe oryzae RNA virus 1(MoRV1)	First_Contig2	ON791638	3165	Magnaporthe oryzae RNA virus	96.809	unclassified
Magnaporthe oryzae mononegaambi virus 2	First_Contig100	ON791636	3681	Magnaporthe oryzae mononegaambi virus 1	84.281	unclassified
Magnaporthe oryzae mymonavirus 2(MoMV2)	First_Contig46	ON791637	4224	Magnaporthe oryzae mymonavirus 1	82.787	unclassified
Rhizoctonia solani dsRNA virus 3(RsRV3)	Contig989	NC032150	502	Rhizoctonia solani dsRNA virus 3	100	*Partitiviridae*
Rhizoctonia solani dsRNA virus 2(RsRV2)	Contig990	MZ043917	712	Rhizoctonia solani dsRNA virus 2	100	*Partitiviridae*
Sclerotinia sclerotiorum hypovirus 2(SsHV2)	Contig995	KF898354	628	Sclerotinia sclerotiorum hypovirus 2	100	*Hypoviridae*
Rhizoctonia fumigata mycovirus	Contig1079	NC_026954	515	Rhizoctonia fumigata mycovirus	100	unclassified
Rhizoctonia solani RNA virus HN008	Contig1208	NC_027529	4437	Rhizoctonia solani RNA virus HN008	100	unclassified
Ustilaginoidea virens botourmiavirus 6(UvBV6)	Contig25222	ON791661	219	Soybean leaf-associated ourmiavirus 2	58.3	*Botourmiaviridae*
Ustilaginoidea virens botourmiavirus 4(UvBV4)	Contig293	ON791659	2331	Sclerotinia sclerotiorum ourmia-like virus 2	51.1	*Botourmiaviridae*
Ustilaginoidea virens botourmiavirus 5(UvBV5)	First_Contig57	ON791660	557	Soybean leaf-associated ourmiavirus 1	50.8	*Botourmiaviridae*
Ustilaginoidea virens botourmiavirus 2(UvBV2)	First_Contig29	ON791657	2496	Rhizoctonia solani ourmia-like virus 1 RNA 1	35.8	*Botourmiaviridae*
Ustilaginoidea virens botourmiavirus 1(UvBV1)	Contig255	ON791656	2314	Magnaporthe oryzae ourmia-like virus	35.6	*Botourmiaviridae*
Ustilaginoidea virens botourmiavirus 3(UvBV3)	Contig1333	ON791658	2268	Sclerotinia sclerotiorum ourmia-like virus 1 RNA 1	28.6	*Botourmiaviridae*
Ustilaginoidea virens mitovirus 1(UvMV1)	First_Contig59	ON791662	2570	Sclerotinia sclerotiorum mitovirus 6	47.2	*Mitoviridae*
Ustilaginoidea virens narnavirus 3(UvNV3)	Contig731	ON791665	2398	Sanya narnavirus 7	72.3	*Narnaviridae*
Ustilaginoidea virens narnavirus 2(UvNV2)	Contig553	ON791664	2395	Plasmopara viticola lesion associated narnavirus 43	67.05	*Narnaviridae*
Ustilaginoidea virens narnavirus 1(UvNV1)	First_Contig93	ON791663	1963	Fusarium poae narnavirus 1	35.4	*Narnaviridae*
Ustilaginoidea virens partitivirus 3A(UvPV3A)	First_Contig43	ON791676	1134	Ustilaginoidea virens partitivirus 3	98.9	*Partitiviridae*
Ustilaginoidea virens partitivirus 2A(UvPV2A)	First_Contig80	ON791675	2018	Ustilaginoidea virens partitivirus 2	98.4	*Partitiviridae*
Ustilaginoidea virens partitivirus 11(UvNV11)	First_Contig14	ON791673	3270	Ustilaginoidea virens nonsegmented virus 1	97.2	*Partitiviridae*
Ustilaginoidea virens partitivirus 6(UvNV6)	Contig1995	ON791668	354	Botryotinia fuckeliana partitivirus 1	94.9	*Partitiviridae*
Ustilaginoidea virens partitivirus 10(UvNV10)	Contig294	ON791672	379	Penicillium aurantiogriseum partitivirus 1	93.2	*Partitiviridae*
Ustilaginoidea virens partitivirus 12(UvNV12)	Contig88	ON791674	1823	Ustilaginoidea virens partitivirus	88.5	*Partitiviridae*
Ustilaginoidea virens partitivirus 8(UvNV8)	First_Contig46	ON791670	661	Colletotrichum truncatum partitivirus 1	85.9	*Partitiviridae*
Ustilaginoidea virens partitivirus 9(UvNV9)	Contig439	ON791671	482	Discula destructiva virus 1	83.3	*Partitiviridae*
Ustilaginoidea virens partitivirus 13(UvPV13)	Contig6891	ON791677	300	Ustilaginoidea virens RNA virus	80	*Partitiviridae*
Ustilaginoidea virens partitivirus 7(UvNV7)	Contig337	ON791669	588	Colletotrichum partitivirus 1	76.6	*Partitiviridae*
Ustilaginoidea virens partitivirus 5(UvNV5)	Contig711	ON791667	1810	Botryosphaeria dothidea partitivirus 1	74.3	*Partitiviridae*
Ustilaginoidea virens partitivirus 14(UvPV14)	Contig321	ON791678	1573	Verticillium dahliae partitivirus 1	74.1	*Partitiviridae*
Ustilaginoidea virens partitivirus 4(UvNV4)	Contig256	ON791666	1625	Botryosphaeria dothidea partitivirus 1	58.1	*Partitiviridae*
Ustilaginoidea virens polymycovirus 1(UvPMV1)	Contig6818	ON791679	1957	Cladosporium cladosporioides virus 1	58.7	*Polymycoviridae*
Ustilaginoidea virens RNA virus M-A(UvRVMA)	First_Contig38	ON791647	3028	Ustilaginoidea virens RNA virus M	99.4	*Totiviridae*
Ustilaginoidea virens RNA virus 6-A(UvRV6A)	First_Contig22	ON791649	2157	Ustilaginoidea virens RNA virus 6	96	*Totiviridae*
Ustilaginoidea virens RNA virus 15(UvRV15)	Contig392	ON791646	4627	Ustilaginoidea virens RNA virus L	81.9	*Totiviridae*
Ustilaginoidea virens RNA virus 13(UvRV13)	Contig28	ON791644	5227	Ustilaginoidea virens RNA virus 1	76.7	*Totiviridae*
Ustilaginoidea virens RNA virus 14(UvRV14)	First_Contig78	ON791645	2507	Ustilaginoidea virens RNA virus 3	75.2	*Totiviridae*
Ustilaginoidea virens RNA virus 11(UvRV11)	Contig21259	ON791642	250	Phomopsis vexans RNA virus	50	*Totiviridae*
Ustilaginoidea virens RNA virus 7(UvRV7)	First_Contig27	ON791648	4661	Ustilaginoidea virens RNA virus 5	49.1	*Totiviridae*
Ustilaginoidea virens RNA virus 10(UvRV10)	Contig94	ON791641	910	Penicillium aurantiogriseum totivirus 1	47.1	*Totiviridae*
Ustilaginoidea virens RNA virus 8(UvRV8)	Contig3788	ON791639	710	Beauveria bassiana victorivirus 1	45	*Totiviridae*
Ustilaginoidea virens RNA virus 12(UvRV12)	First_Contig136	ON791643	939	Rosellinia necatrix victorivirus 1	37.9	*Totiviridae*
Ustilaginoidea virens RNA virus 9(UvRV9)	First_Contig40	ON791640	4956	Gremmeniella abietina RNA virus L1	35.9	*Totiviridae*
Ustilaginoidea virens unassigned RNA virus HNND-1-A	Contig508	ON791651	2912	Ustilaginoidea virens unassigned RNA virus HNND-1	97.1	unclassified
Ustilaginoidea virens mycovirus 1	First_Contig11	ON791650	1149	Ustilaginoidea virens mycovirus	94.9	unclassified
Ustilaginoidea virens virus 1(UvV1)	Contig21350	ON791655	533	Alternaria tenuissima virus	68.9	unclassified
Ustilaginoidea virens narna-like virus 3(UvNLV3)	Contig1170	ON791654	3128	Beihai narna-like virus 23	37	unclassified
Ustilaginoidea virens narna-like virus 1(UvNLV1)	Second_Contig2	ON791652	3099	Beihai narna-like virus 21	34.3	unclassified
Ustilaginoidea virens narna-like virus 2(UvNLV2)	First_Contig75	ON791653	689	Beihai narna-like virus 22	27.1	unclassified

## Data Availability

The sequences reported in the present manuscript have been deposited in SRA with accession numbers PRJNA845944.

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
