# Peer review of "Metatranscriptomic Analysis Reveals Rich Mycoviral Diversity in Three Major Fungal Pathogens of Rice"

_ijms, 2022, doi:10.3390/ijms23169192_

Round 1

Reviewer 1 Report

Zhenrui et al. are presenting an interesting study of using metatranscriptomic  analysis to reveal the mycoviral diversity in three major fungal pathogens of rice. Overall, the manuscript is well-organized with detailed description of the experimental details. The outcome and the conclusion is supported by the data observation. The following are specific points.

1. Figure 2A uses a pie plot to show the number/distribution of mycoviral species in each viral family. It will be beneficial to add the percentage or the number to each colored region to provide an intuitional idea to readers.

2. In table 1, and the section of "2.3. Sequences related to members of the families Botourmiaviridae and Hypoviridae" the amino acid sequence identity was used. However, in the section of "2.6. Twelve Predicted Novel Viruses in the Family Totiviridae" the sequence similarity was used instead. The reviewer understand that similar residues may have similar properties. But what is the rationale behind using identity and similarity inconsistently across the manuscript?

3. Authors mentioned that "this analysis identified a wide diversity of previously described and novel mycoviral species, some of which could result in less growth rate, the decline of pathogenicity and other phenotypes of their fungal hosts." Is it possible to associate species with symptoms mentioned above in the phylogenetic tree?

4. In phylogenetic trees, blue, red, and yellow colors denote viruses in U. virens, P. oryzae, and R. solani. For viruses without any color denotation, it is not clear if they are just not detected in this study or they have not yet been detected to infect rice. A clarification is appreciated.

Overall, the reviewer would suggest a minor revision.

Reviewer 2 Report

The revised manuscript describes mycoviruses accompanying 3 main fungal pathogens of rice. The topic is important as any pathogen can negatively affect any crops. For that reason recognition of any natural anty-fungal agents is important and useful to prevent further crops diseases. As such studies requires modern molecular tools I can state that the manuscript falls into the scope of IJMS.

It is well organized with informative title, good abstract and keywords. Introduction is good providing good background, the rationale of conducted study and clear aim.

There are corrections incorporated of a probably previous revision. The final version of the manuscript should prepared for the final production. The methods are well described and supported by two tables.

I noticed that Figures 1 and 2 don't fit to the page - they are partially visible. I wonder if the fonts will be readable after figures downscaling. Besides this the results describes well obtained data and are supported with graphs. Discussion is well written too.

Author Response

This manuscript is a resubmission of an earlier submission. The following is a list of the peer review reports and author responses from that submission.

Round 1

Reviewer 1 Report

Manuscript "Metatranscriptomic analysis reveals rich mycoviral diversity in three major fungal pathogens of rice" by Zhenrui He, Xiaotong Huang, Yu Fan, Mei Yang, Erxun Zhou representing study of the three major fungal patho-gens of rice, Pyricularia oryzae, Ustilaginoidea virens and Rhizoctonia solani, sampled in southern China. A detailed analysis of microviruses was performed with partial identification and division into groups. The work is complete and interesting. Small inaccuracies in the design should be corrected. The drawings are excessively small, the inscriptions are indistinguishable. The work can be accepted with the correction of this comment.

Reviewer 3 Report

Zhenrui He and colleagues present an article on the identification and initial classification of novel sequences of viral origin using RNA sequencing and primary bioinformatic analysis. The manuscript is carefully and comprehensibly written and the results appear to be interesting and might prove important for relevant studies in this field.

Given the metatranscriptomic nature of the manuscript, authors should consider presenting their data in a more "impressive" way, also by including additional bioinformatic analyses, in order to further highlight the results, better support the conclusions and thus attract more attention to your manuscript.

Figure 1: Abnormal colony morphology can only be compared with the wt. Please include images of wt strains for these genera.

Lines 155-158: These results, or at least part of those, should be included along with the primers as supplementary material.

Line 442: Please provide an analysis summarizing and correlating strains with hypovirulent characteristics with the identified viruses.

The part in the Material and Methods referring to metatranscriptomic sequencing should be described in more detail.

Line 285: What do the authors mean in this case with “unpublished data”? Aren’t the raw sequencing data not provided in the sequences to be deposited? If not, then the relevant information should be omitted.

Please go through the manuscript and correct minor typos and errors, like in Lines 61 and 324.

Round 2

Reviewer 3 Report

The authors have responded succesfully to most comments and suggestions. Nevertheless, I still encourage authors to try and better highlight their interesting results in future studies.